# Systemic neurotransmitter responses to clinically approved and experimental neuropsychiatric drugs

Hamid R. Noori[1,2,3,5], Lewis H. Mervin[4], Vahid Bokharaie[2], Özlem Durmus[1], Lisamon Egenrieder[1], Stefan Fritze[1], Britta Gruhlke[1], Giulia Reinhardt[1], Hans-Hendrik Schabel[1], Sabine Staudenmaier[1], Nikos K. Logothetis[2], Andreas Bender[4] & Rainer Spanagel[1]

Neuropsychiatric disorders are the third leading cause of global disease burden. Current pharmacological treatment for these disorders is inadequate, with often insufficient efficacy and undesirable side effects. One reason for this is that the links between molecular drug action and neurobehavioral drug effects are elusive. We use a big data approach from the neurotransmitter response patterns of 258 different neuropsychiatric drugs in rats to address this question. Data from experiments comprising 110,674 rats are presented in the Syphad database [www.syphad.org]. Chemoinformatics analyses of the neurotransmitter responses suggest a mismatch between the current classification of neuropsychiatric drugs and spatiotemporal neurostransmitter response patterns at the systems level. In contrast, predicted drug–target interactions reflect more appropriately brain region related neurotransmitter response. In conclusion the neurobiological mechanism of neuropsychiatric drugs are not well reflected by their current classification or their chemical similarity, but can be better captured by molecular drug–target interactions.

[1] Institute of Psychopharmacology, Central Institute of Mental Health, Medical Faculty Mannheim, University of Heidelberg, J5 68159 Mannheim, Germany. [2] Max Planck Institute for Biological Cybernetics, Max Planck Ring 8, 72076 Tübingen, Germany. [3] Courant Institute for Mathematical Sciences, New York University, 251 Mercer Street, New York, NY 10012, USA. [4] Centre for Molecular Informatics, Department of Chemistry, University of Cambridge, Lensfield Road, Cambridge CB2 1EW, UK. [5] Present address: Neuronal Convergence Group, Max Planck Institute for Biological Cybernetics, Max Planck Ring 8, 72076 Tübingen, Germany. These authors jointly supervised this work: Hamid R. Noori, Nikos K. Logothetis, Andreas Bender, Rainer Spanagel  Correspondence and requests for materials should be addressed to H.R.N. (email: hamid.noori@tuebingen.mpg.de)

Mental disorders are common throughout the world, yet treatment success is modest. In general, the difficulties in developing optimal therapeutic agents for neuropsychiatric disorders are due to limitations of understanding the pathophysiology of these diseases and consequently lack of appropriate biomarkers and molecular targets[1]. Modern psychiatry therefore searches for new ways of classifying psychopathology based on dimensions of observable behaviour and neurobiological measures to improve diagnosis and treatment[2].

Currently, the classification of neuropsychiatric drugs is primarily based on their clinical and therapeutic effects. Since 1996, the World Health Organization emphasizes the use of the anatomical therapeutic chemical (ATC) classification system for drugs as an international standard. In the ATC, medications for psychiatric practice are classified under the central nervous system (CNS) category and are grouped on the basis of their therapeutic use such as antidepressants, mood stabilizers, anxiolytics, sedatives or hypnotics, antipsychotics, psychostimulants, anticonvulsants and anti-dementia drugs. Further classification levels relate to a combination of chemical structure, pharmacological action and/or therapeutic use. One criticism of the ATC system is that agents are not classified and grouped systematically. Hence in many ATC main groups, pharmacological groups have been assigned allowing drugs with several therapeutic uses to be included without specifying the main indication. Furthermore, for psychiatry the ATC system relies on largely subjective syndrome-based disease classifications described in either the fifth edition of *Diagnostic and Statistical Manual of Mental Disorders* (DSM-5) or the 11th version of the *International Statistical Classification of Diseases* (ICD-11). An intrinsic feature of psychiatric diagnoses is the presence of overlapping criteria and comorbidities. A good example is the diagnosis of anxiety and personality disorders. In the majority of cases, patients with an anxiety (or a personality) disorder fulfil criteria for at least one additional anxiety (or personality) disorder[3]. The ATC system inherits this feature in a natural way. For example, selective serotonin reuptake inhibitors (SSRIs), which present the most commonly used antidepressants, can be prescribed for anxiety disorders[4] as well, while second-generation antipsychotics such as risperidone are effective in treatment of depression and anxiety[5].

Clearly, a classification system for neuropsychiatric medications that is based on the pharmacological and molecular action and the resulting neurobehavioral changes would be more appropriate. Many compounds interact with multiple targets leading to more complex modes of action[6,7]. Thereby, the localized molecular effects are not transformed to higher spatio-temporal scales in a trivial manner and potentiating effects at the molecular level may diminish or even have inhibitory impact on the network systems level. Moreover, the causal links between the complex multivariable molecular processes modulated by a drug and the resulting neurobehavioral effects are largely not understood. Thus, a focus on molecular modes of action by receptor pharmacology can only go so far in explaining drug effects on CNS, given it does not fully consider multiscale effects on brain biology[8]. Several biological and chemical databases for therapeutic and experimental drugs have been constructed. In particular, databases such as the National Institute of Mental Health Psychoactive Drug Screening Programme[9], Receptoromics[10], Drug Voyager[11], PubChem[12], Ligand Expo[13], ZINC[14], STITCH[15] and KEGG DRUG[16] have been developed that integrate diverse information such as compound structures, drug targets, and molecular pathways modulated in a biological system. While these databases provide useful information for drug discovery and repurposing processes, they focus on the chemical and molecular level (i.e. drug A binds to receptor B) and also do not address how the molecular drug effects relate to the diverse multi-dimensional neurobehavioral changes observed on the organism level.

Hence, using multimodal dimensions related to pharmacological and clinical domains and molecular modes of action, a taskforce composed by experts from different societies on Neuropsychopharmacology has developed a modified system, the so-called *Neuroscience-based Nomenclature*[17], to replace indication-based classifications such as ATC.

Here we provide a novel evidence-based characterization of neuropsychiatric drugs at a systems level. On the systems level of neurotransmitters we have integrated all published information on the spatio-dynamical changes in neurochemistry as measured by microdialysis following acute drug application in rats. In vivo microdialysis is a crucial method to characterize the quantity neurotransmitters and their metabolites, neuropeptides and hormones within interstitial tissue fluids[18] following different pharmacological manipulations[19], and as such reflects very well the spatio-dynamical changes in neurochemistry following acute drug application. We present all extracted data in a large database, Systematic Pharmacological Database or *Syphad*, and use a set of chemoinformatics tools[20,21] with which causal links between the polypharmacology of neuropsychiatric drugs and their effects at systems level are semi-quantitatively established.

## Results

**The *Syphad* database summarizes neurochemical responses of neuropsychiatric drugs.** Systematic literature search identified the neurochemical response patterns that represent drug-induced changes in extracellular concentrations of 59 neurotransmitters, modulators, neuropeptides and metabolites within a network of 117 brain regions stretched over both hemispheres. In total, neurochemical response data from 258 clinically approved and experimental neuropsychiatric are provided in an open-access online platform called *Systematic Pharmacological Database* or *Syphad* [www.syphad.com]. The data was retrieved using automatic keyword-based search (with a search string length of 360 keywords and 13,608 keyword combinations) and manual grey search on electronic databases. In the first search step 214,288 abstracts, titles, or both were identified from original publications. Out of these, 15,777 studies were relevant for data mining and data from 3383 original research articles on in vivo microdialysis in rat brain (covering studies involving 110,674 rats) were selected for the meta-analyses and subsequent database creation (Fig. 1).

Syphad contains 10,510 unique 18-dimensional vector numerical and nominal entries (Supplementary Data). Thereby, each vector contains data in five categories. The first category contains the Pubmed identification number (PMID) as well as the number of animals used in a particular published study, where the PMID uniquely assigns every vector to the original publication and where the number of animals is a measure for the robustness of experimental observations and thereby serves as the weight for subsequent meta-analysis (see methods section). The second category contains biologically relevant data, including strain, sex, age, state of consciousness, i.e., awake/freely moving or anesthetized (and if so agent and dose), brain region and neurotransmitter system. The third category relates to the microdialysis setup parameters, namely perfusate, its calcium concentration and the flow rate of the perfusion. The fourth category contains the drug designation, dose of the drug as well as route of administration. Thereby, this data category represents the experimental input, while categories 2 and 3 are the model covariates. The fifth category is the experimental output by means of dynamics and magnitude, i.e., the time-point at which the drug

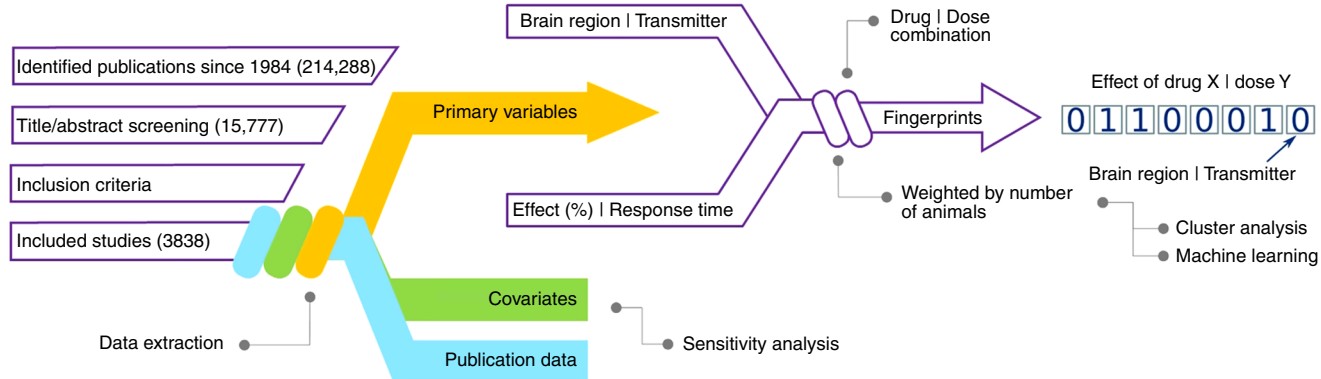

**Fig. 1** Flowchart of data mining, extraction and fingerprint generation procedures. Keyword-based search of online databases and manual grey search of literature identified 214,288 publications since 1984, out of which 15,777 were screened for content. Three categories of data were extracted from 3838 studies that fulfilled the inclusion criteria (microdialysis AND rat AND brain AND systemic drug administration). For each drug, dose pairing, the normalized effect was calculated as a meta-analysis weighted by the number of animals used in each study and the robustness of the values were estimated by sensitivity analysis with respect to covariates such as age, strain, sex and microdialysis parameters. The data was then transformed into bit arrays which represent the neurochemical response fingerprints within the neurochemical connectome of the rat brain

induces its maximum effect (peak time) and the effect size normalized to baseline concentration levels (peak%BL).

**Data distribution in *Syphad*.** In all, 40 out of the 59 neurotransmitters, modulators, neuropeptides and metabolites in Syphad have at least five unique entries, which in total represent 99.6% of all entries. Thus, the remaining 0.4% of all entries represent 19 neurochemicals in the database. This underrepresentation of a large proportion of neurochemicals (mostly large peptides) is explained by technical limitations of the microdialysis technique and the adjunct detection method. Especially large neuropeptides such as substance P, corticotropin-releasing hormone, ß-endorphin, dynorphin or somatostatin are extremely difficult to detect[22] and therefore fall into this category.

The vast majority of included studies have collected data related to monoamines and their metabolites. Thus in Syphad, 78.7% of the unique entries relate to monoamines and their metabolites (dopamine 38.1%, 5-HT 18.7% and their main metabolites DOPAC 10.3%, HVA 7.6%, and 5-HIAA 4.0%). Thereby, a total of 56.4% of entries relate to both, measurements in striatum (3670 in absolute terms) and nucleus accumbens (3150). In contrast, noticeably fewer studies reported measurements of the main excitatory and inhibitory transmitter systems (acetylcholine 6.3%, glutamate 3.4% and GABA 1.9%). This observation demonstrates a general skewness of microdialysis studies that is also reflected in the Syphad database. Included studies date back as early as 1984. 96% of the database entries relate to male, 80% to adult and 89% to freely moving animals, while 60% and 30% of entries provide data for Sprague-Dawley and Wistar rats, respectively.

**Database sensitivity analysis.** Sensitivity analyses with respect to covariates were performed to ensure the robustness of the meta-analysis. For 5.4% of drugs in the database (i.e. 14 out of 258), microdialysis experiments were conducted using female animals and in 1.9% of cases, a statistical analysis of sex as a covariate was possible. For amphetamine 0.2–5 mg/kg ($p > 0.05$ for all doses), apomorphine 1 mg/kg ($p = 0.49$), cocaine 10 mg/kg ($p = 0.99$), methamphetamine 3 mg/kg ($p = 0.66$) and venlafaxine 20 mg/kg ($p = 0.29$) one-way ANOVA did not show any significant differences between male and female animals. Apart from a few exceptions, there were no systematic differences in drug-induced changes in neurotransmitter

concentrations with respect to other biological covariates, age and strain. Risperidone (0.1 mg/kg) increased dopamine concentration in prefrontal cortex in adolescent rats by 170% larger than adult animals ($p = 0.0003$, one-way ANOVA). Administration of 10 mg/kg cocaine increased dopamine concentration in adolescent animals by 306 ± 3% and thus, significantly less ($p < 0.05$, one-way ANOVA) than in adult rats (371 ± 0.2%). Strain as a covariate did not affect the robustness of meta-analyses but in a few exceptions (0.9% of entries). Different doses of clozapine affected dopamine levels in prefrontal cortex and striatum in a nonlinear manner in both Sprague-Dawley (SD) and Wistar animals; however, the one-way ANOVA suggests that changes induced by doses of 10 and 20 mg/kg were more pronounced in Wistar than SD rats ($p < 0.01$). In comparison with SD rats, 3 mg/kg of paroxetine induced a twofold stronger increase in 5-HT concentrations in the frontal cortex of Wistar rats ($p < 0.05$). However, dopamine metabolites (DOPAC and HVA) in nucleus accumbens in response to 5 mg/kg morphine, but not other doses ($p < 0.01$), and to 5-HT in the frontal cortex in response to 10 fluoxetine ($p < 0.0001$) in SD rats were enhanced significantly more than in Wistar rats.

We finally analysed the reproducibility of neurochemical response assays in the database by correlating experiments of identical conditions (that is, measured for the same drug, transmitter, region, dose, route of administration and time parameters). To do this, response measurements were converted to 1 or 0 for up- or downregulation (above or below the 100% baseline, respectively), and the standard deviation was subsequently calculated on a per-compound basis. Overall 98.8% of experiments resulted in the same response vector with an average of 0.13 ± 0.08 between the standard deviations across compounds, and a median of 0.0, which hence indicates that there is strong reproducibility (considering up/down regulation) across microdialysis assays and provides confidence to conduct further analyses with this database.

**ATC codes and neurochemical response correlate only weakly.** We first investigated whether ATC classifications and neurochemical response patterns in different brain regions were correlated, and if so, to which extent the current classification has a sound neurochemical basis. This analysis compares the neurochemical response patterns of compounds extracted from *Syphad* in the form of a bit arrays (as described above—represented by 1

or 0 bits, respectively), which are described in detail within the Data Transformation section in Methods. The bit array representations of response patterns were generated for 1813 experimental measurements covering 44 distinct brain regions and 59 neurochemical components, of which 1034 (~57%) and 799 (~43%) of the measurements are considered upregulated and downregulated (above 100% or below the baseline control), respectively. Based on the Tanimoto coefficient (Tc) similarity calculated for drug-induced neurochemical alterations, our findings (Fig. 2a) show whether compounds with similar codes more often exhibit similar neurochemical response patterns (intra-ATC code similarity) compared to other compounds across other ATC classifications (inter-ATC code similarity). Tanimoto similarity is commonly used in the cheminformatics field for compound fingerprint-based similarity calculations[23,24], where a maximum score of 1.0 represents two compounds with complete overlap

between their (shared) experimental neurochemical-brain response profiles, and a score of 0.0 represents two compounds with no overlap. Thereby, the term fingerprint stands for the above-mentioned bit array representation of the neurotransmitter response pattern. The results comprise 9688 and 19,736 intra- and inter-similarity comparisons, respectively, and illustrate (Fig. 2a) that compounds within ATC classes show a higher median of Tc similarity for neurochemical-brain response patterns, of ~0.43, compared to compounds between ATC classes, where the median similarity is ~0.33. The two comparisons, however, consist of many extreme values, as outlined by their stretched u-shape distribution. A two-sided Kolmogorov–Smirnov test gave a $p$-value <0.001 (6.31$^{e-56}$) showing that the two sets of similarities are significantly different, which indicates that ATC codes indeed do correlate with compound mechanism in terms of neurochemical response to a

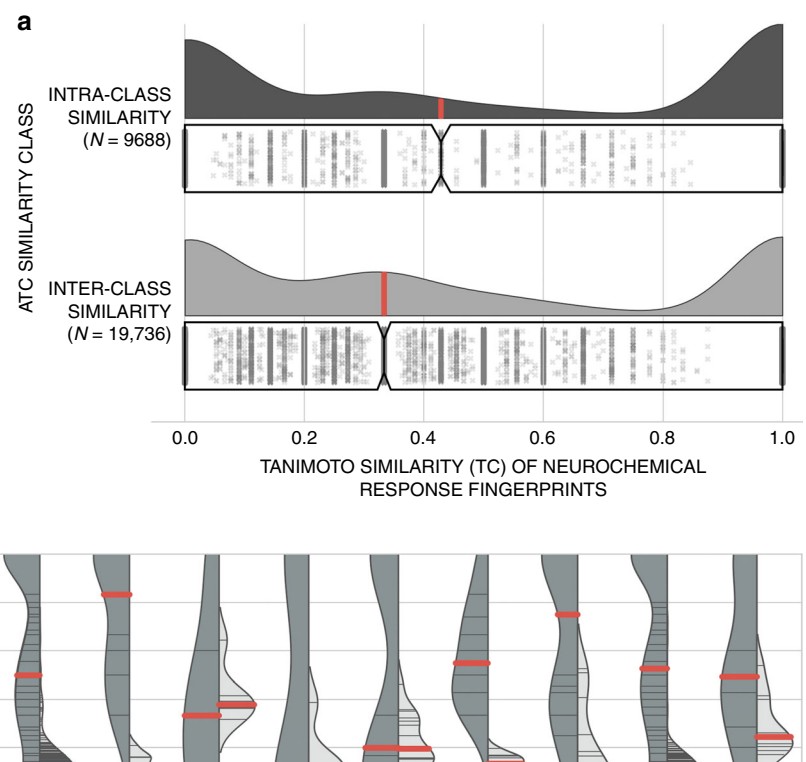

**Fig. 2** Neurochemical similarity analysis. **a** Intra- and Inter-ATC code similarity of neurochemical response patterns across brain regions. Distributions are shown using kernel density estimation (KDE) with median similarities identified by the red marker and via the underlying boxplot with the notch denoting median, the outer boxes denoting the lower and upper-quartile ranges and the jitter scatter plot outlining the underlying observations. Compounds within the same ATC classification (intra-similarity) exhibit higher median similarity in neurochemical response than between ATC codes (inter-similarity), with a median response fingerprint Tanimoto coefficient's of 0.43 and 0.33, respectively. The underlying distributions are statistically different with a two-sided Kolmogorov–Smirnov test $p$-value of 6.31$^{e-56}$. However, the similarity distributions are quite wide (interquartile ranges span 0.0 to 1.0 for both intra- and inter-similarities), with large overlap. Hence, ATC codes could also be considered to be not very meaningful descriptors of neurotransmitter activities. **b** Intra-ATC class similarity of the neurochemical and chemical fingerprints of compounds for ATC classes with 4 or more representative compounds. The number of compounds underlying each distribution is denoted using ($N =$), with the underlying distribution shown using KDE, the median shown via a solid red line and the underlying observations shown via the black stick markers. Our results highlight there are significant differences between different ATC classifications. N.B.: Only ATC classifications with $N \geq 4$ (~3% of drugs in the database) are shown. The so-called combined subset category shows the distribution of the combined subset across the eight ATC classes shown and number of distinct compounds

statistically significant extent. However, the wide distribution range of the two similarities suggest that this finding is not robust. With standard deviations of ~0.42 and ~0.45 for intra- and inter-class similarities, respectively, and a significant number of compound pairs from the same ATC class showing no similarity on the neurotransmitter response level whatsoever, ATC codes seem not to capture the neurochemical effects of drugs in all cases.

Furthermore, we conducted a sensitivity analysis to investigate the robustness of the similarity analysis to characterize the impact of any bias towards certain ATC codes towards the overall distribution. Combinatorial exclusion of ATC codes induces a standard deviation of <0.01 and 0.02 between the median inter- and intra-class similarities, which suggests robustness of this intra- and inter-class similarity analysis.

**Chemical structure and transmitter changes correlate weakly.** We next investigated whether chemical structure and neurochemical response are more conserved within ATC classes, which to an extent would be suspected, both due to related modes of action and the tendency of pharmaceutical industry to generate so-called me-too drugs[25]. Figure 2b outlines the results from our analysis when aggregating the previous neurochemical response profiles by ATC codes with four or more representative compounds and contrasting these distributions with the similarity of compounds using chemical structural descriptors, namely extended connectivity fingerprints (ECFP_4[24]). Eight ATC codes included sufficient compounds, a subset of which comprises 58 distinct compounds providing 452 similarity comparisons. There are generally significant differences between neurochemical and chemical spaces across ATC classifications (the 'Combined subset' column), although this distribution differs significantly between ATC classes.

One class where neurochemical responses are rather similar, while chemical structures differ widely, is ATC code A08A (antiobesity preparations). For this classification we found the highest intra-class neurochemical response similarity (median Tanimoto coefficient of ~0.82), while compounds were still exhibiting among the lowest similarity in structural fingerprint (bit array representation) space (median Tanimoto coefficient of ~0.1). Hence, similar neurochemical response does not generally imply similar chemical structure.

This applies also to the class of antipsychotics drugs (N05A), which shows a neurochemical response similarity with a high median Tanimoto coefficient of ~0.52, but low chemical structure similarity with a median Tanimoto coefficient of ~0.18. This finding is not surprising on a *target* level when considering that for the last half-century, almost all approved antipsychotic drugs have affinity for the dopamine D2 receptor as an apparently essential aspect of their mechanism of action, and also due to the biased (me-too) nature of antipsychotic medicine discovery[26]. However, the apparent diversity of modes of action on the neurochemical level in this compound class (represented by the wide distribution and median Tanimoto coefficient of ~0.52) is far more diverse than the simple requirement of activity on the D2 receptor would suggest, a finding which is not apparent from the protein-based activity definition.

Other examples for large mismatches between neurochemical response similarity and chemical structure similarity relate to the classes of hypnotics and sedatives (N05C), with the second highest neurochemical response fingerprint of ~0.75 vs. the lowest median chemical response fingerprint of ~0.1. Antidepressants (N06A) also show large differences in the ranking of neurochemical and chemical spaces (with median Tanimoto coefficient ~0.5 vs. ~0.13) along with psychostimulants (N06B) (median Tanimoto coefficient of ~0.5 vs. ~0.22) (Fig. 2b).

Thereby, analysis of the inter-class similarity between the neurochemical response and chemical structures suggests that inter-class compounds are generally dissimilar in both neurochemical response and chemical structure similarity (overall median Tanimoto coefficients of ~0.33 and ~0.10, respectively). Thus, chemical similarity and/or identical ATC use classes do not relate to the action of drugs in an identical or similar manner on the neurochemical level (and vice versa).

**Frequency of neurotransmitter-brain region pair response.** The mode of action of a drug is not only determined by its change in a particular neurotransmitter, but also by the brain sites the neurochemical events take place; here referred to as the spatial neurochemical response pattern. Thus by calculating the fraction of times compounds administered within the same brain regions elicit up- or downregulation of neurochemical response, we investigated whether particular brain regions show more frequent neurochemical responses to drug treatment than others. The results of the analysis, displayed in Fig. 3, suggests that neurotransmitters such as dopamine (DA) have a high frequency of upregulation across different brain regions, where 21 of the 26 measurements (~80%) exhibited a fraction of upregulation above 50%. In other words, dopamine responds frequently and largely in a non-specific manner with respect to applied drug or measured brain regions. This result can be contrasted with GABA, which has a higher frequency of downregulation across 9 of the 14 brain regions for which data is available.

In order to illustrate this point on an individual compound level, hierarchical clustering of compound activity across brain region and neurotransmitters was performed (Fig. 4 & Supplementary Fig. 1). The analysis suggests that drugs from the same ATC class rarely cluster, illustrating that ATC class and changes in neurotransmitter levels across different brain regions are only very weakly correlated. One prominent example relates to the selective serotonin reuptake inhibitors paroxetine and citalopram (ATC codes of N06A) that separate into two distinct branches of the dendrogram. This indicates that despite their similarities in clinical use[27,28] and molecular modes of action, there are significant differences with respect to their effects at the brain region and neurotransmitter level. To an extent, this might be explained by the more selective inhibitory activity of citalopram on serotonin reuptake[27], where paroxetine also affects acetylcholine and noradrenaline reuptake; on the other hand, even the antihypertensive MAO-A inhibitor pargyline is found to be more similar in neurochemical response space to paroxetine than citalopram, which illustrates that ATC codes and effects on spatial neurochemical response patterns do not well agree with to each other in case of this set of compounds.

**Linking drugs with their predicted molecular interactions.** To study the relationship between spatial neurochemical response patterns and key molecular drug–target interactions, we next investigated which bioactivities of a drug against protein targets are more frequently associated with neurotransmitter level changes across brain regions. This analysis is based on in silico protein target predictions[29] for compounds in *Syphad*, where computationally, based on large bioactivity databases, a complete putative ligand-target interaction matrix is generated. Only models trained with rat bioactivity data were used since this is where the experimental data from Syphad is derived, and predictions were only generated for those targets expressed in brain tissue. Complete details on the in silico protein target prediction and model selection are provided in the Methods section on "Compound analysis based on experimental data". Overall predictions were available for 100 in silico rat targets, given the

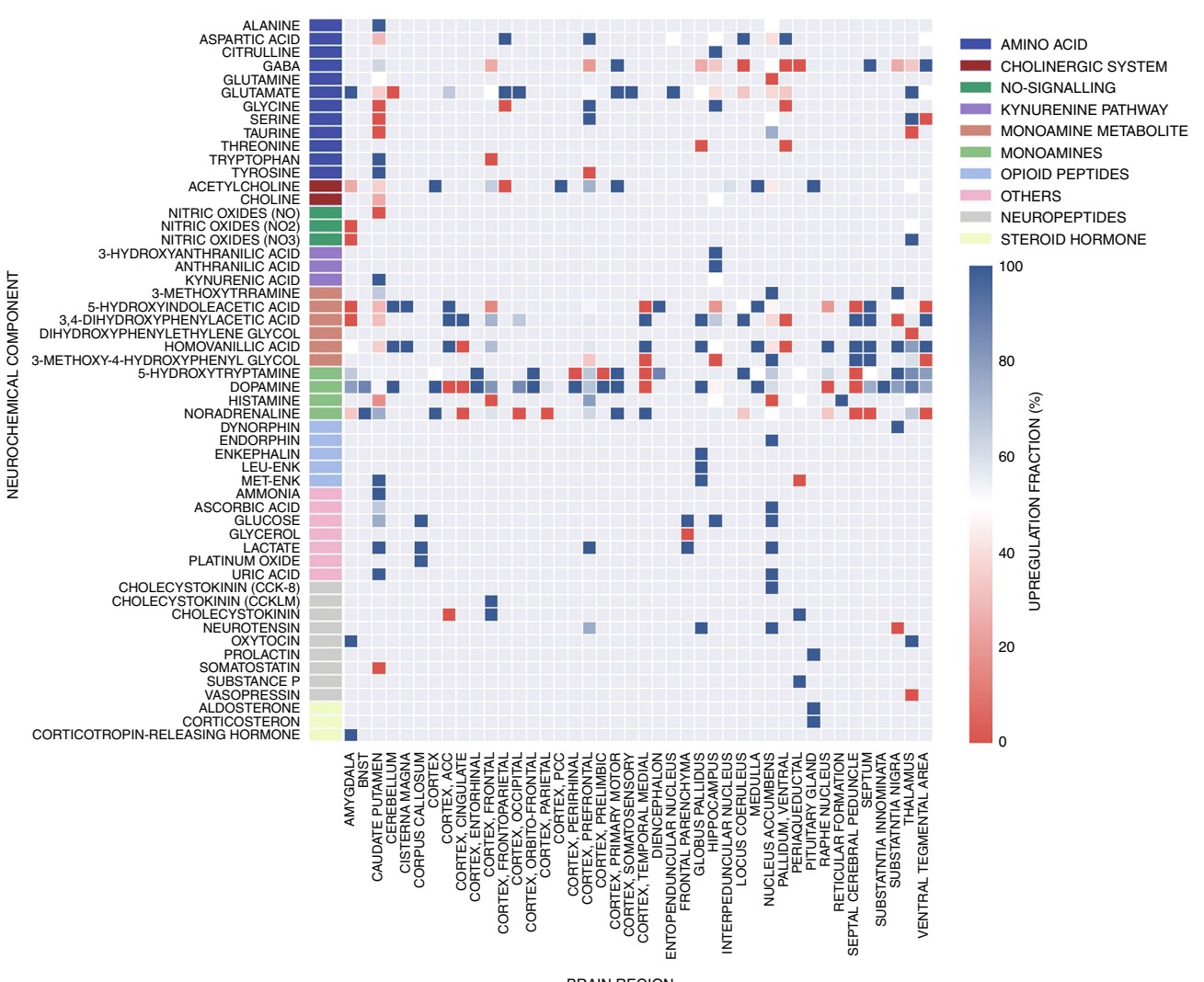

**Fig. 3** Frequency of neurotransmitter-brain region pair response upon compound administration. Our analysis highlights the fraction of times compound administration led to an increased (red) or decreased (blue) neurochemical response across each brain region and neurochemical component pair. The visualization contains a total of 1813 compound measurements available across 44 distinct brain regions. It can be seen that neurochemical components such as dopamine (DA) have high frequency of upregulation across different brain regions, indicating a rather unspecific response of this neurotransmitter to drug treatments. Missing measurements are coloured in grey, which also highlights the sparsity of the neurochemical experiments. Neurochemical components are ordered by classification (indicated via row colouring)

aforementioned filtering, yielding 258 drug × 100 possible drug–target interactions. As to date only ~1.7% of the drug–target interaction matrix have been experimentally determined for the drugs, thereby providing a rationale to conduct this analysis using in silico protocols. The systematic analysis (Fig. 5 & Supplementary Fig. 2) outlines that the proportion of times (averaged percentage hits) that targets were predicted is modulated by compounds eliciting response at corresponding neurochemical components. The most associated targets (ranked by top hit percentages) and their respective neurochemical components from this analysis are presented in Table 1, which can be used as a reference for future experiments to confirm our novel links between targets modulating neurochemical signalling. Twelve of the top 25 identified protein target-neurotransmitter relationships are already supported by the literature, and a further three are identified in the literature as specific candidate genes requiring confirmation. For example, alpha 1 adrenoceptor (ADRA1B) activation potentiates taurine response mediated by protein kinase C in substantia nigra neurons[30]; opioid antagonism of opioid receptor delta 1 (OPRD1) modulates oxytocin levels[31]; and

ligands of muscarinic 4 (CHRM4) and muscarinic 5 (CHRM5) receptors are known to regulate dopamine transmission[32]. The analysis also provides the identification of novel ways of modulating neurotransmitter levels by connecting target and neurotransmission response spaces, such as the predicted link between glutamate ionotropic receptor NMDA type subunits (GRIN2A and GRIN2C, GRIN2D and GRIN1) and the target-dependent regulation of kynurenic acid (KYNA) or serotonin/melatonin precursor tryptophan (TRP). The analysis was further conducted based on the aggregated protein target prediction rates across brain regions (instead of neurotransmitter), highlighting clusters of brain region and protein target tuples, and hence a general correlation between compounds targeting specific proteins more frequently modulating neurochemical response within specific brain regions (Fig. 6). Findings from this analysis can also be used in a similar manner to the previous neurochemical component analysis, that is, to direct future biochemical experiments and inform which microdialysis assays should be performed to corroborate our putative links between targets modulating response within brain regions.

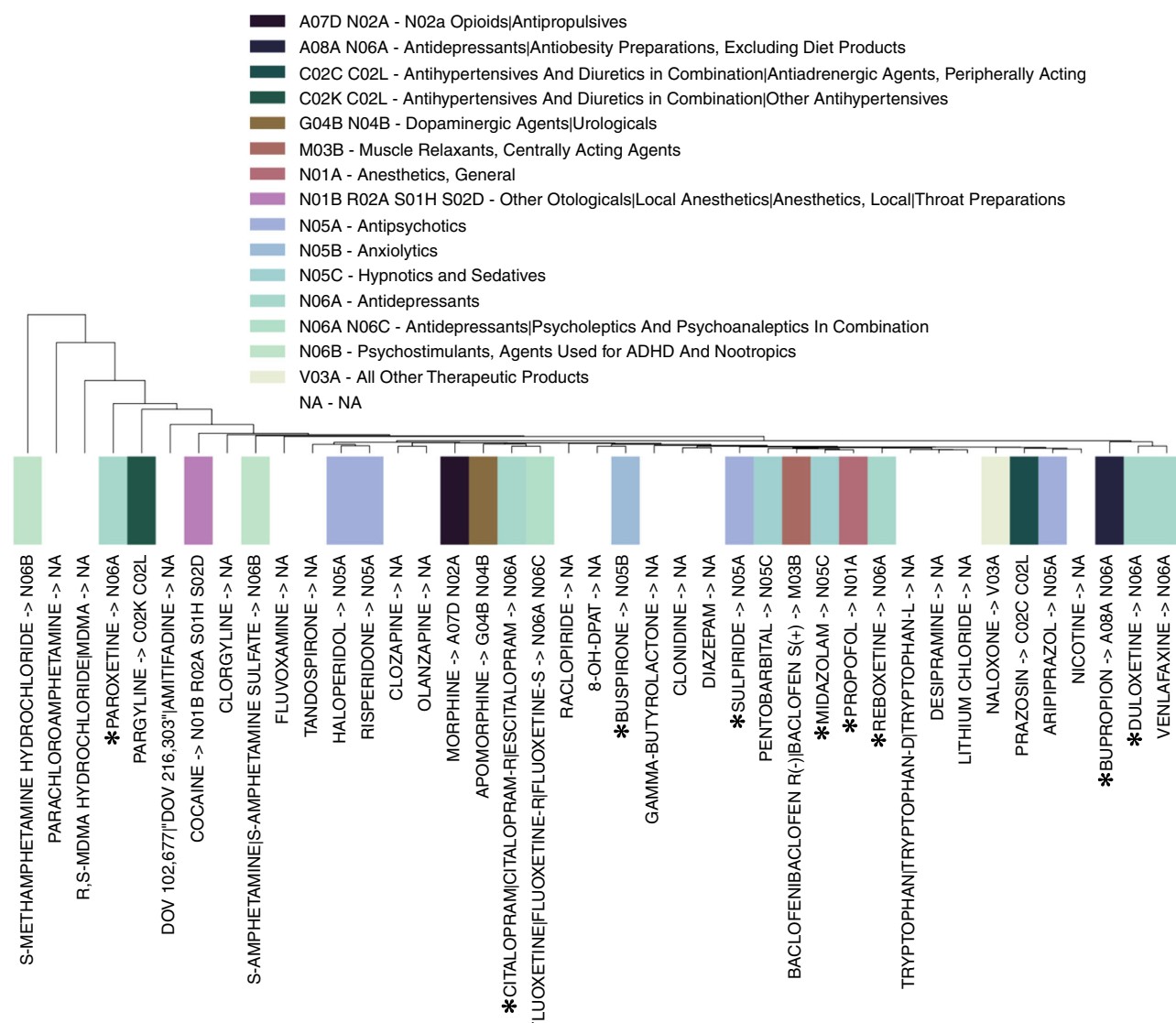

**Fig. 4** Compound activity across brain regions and neurotransmitters. The analysis highlights that the historical classification of active ingredients of drugs (ATC codes) does not cluster well with their actual mode of action. There is little correlation between compound activity and codes which hence indicate that the current classification system is not exhaustive or complete. A subset of the compounds is shown here upon filtering the database for matrix completeness of 50%. ATC codes are shown using colour (key above). Compounds discussed in the main text are outlined using *

**In silico predicted molecular drug–target interactions**. To investigate if the predicted drug–target interactions better reflect brain region related neurochemical response than ATC codes on a quantitative level, we calculated the averaged (mean) degree of mutual information (as outlined in the Methods section "Mutual information analysis") of either ATC codes or predicted protein target spaces with drug-induced changes in extracellular neurotransmitter concentrations. Mutual information is a measure used in bioinformatics to describe the similarity (or dependence) between two features (here either an ATC code or protein annotation versus neurochemical response) in a dataset[33]. A score of 1.0 represents the situation when two features are perfectly dependent (and hence the information about the neurochemical response of a compound can be perfectly derived from either the ATC code or protein target prediction). In turn, a score of 0.0 represents mutual independence between the features. Overall findings averaged across ATC codes and protein targets (shown in Fig. 7) suggest that the two sets of mutual information scores are relatively similar in terms of their median distribution, with scores around ~0.623, which would initially indicate that ATC

codes indeed do correlate with compound mechanism in terms of neurochemical response. However, the distributions of predicted protein target mutual information are wider (standard deviation of 0.010 vs. 0.007), with a significantly larger tail towards higher mutual information scores, achieving values of up to 0.68. Thus, this finding supports the view that certain predicted drug–target interactions are more appropriate indicators of brain region related neurochemical changes.

To outline the robustness of our findings, we analyse the extent of biases towards particular ATC codes or targets which may affect the distribution of mutual information scores. Thereby, we explored the degree to which the median mutual information score obtained is shifted upon leaving each ATC code or target model out of the bit array representations. Our results show there is a standard deviation of <0.01 and 0.01 between the median neurochemical response mutual information scores versus the ATC and protein prediction fingerprints, respectively, and hence the findings are robust towards variations.

Based on this finding, we next analysed the five ATC classes with the highest mutual information (i.e. the most informative

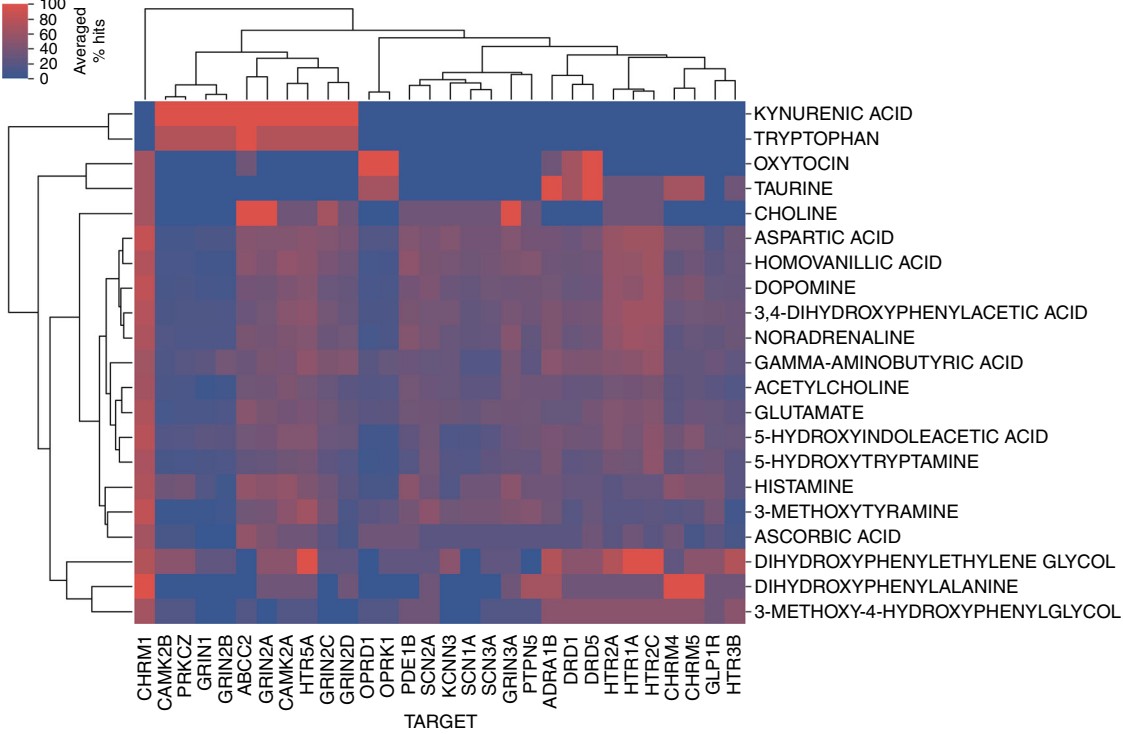

**Fig. 5** Systematic analysis of bioactivities against protein targets associated with neurochemical changes. A larger percentage (red) indicates which protein targets are more often predicted to be targeted by drugs which change a given neurotransmitter level. Target families are shown to frequently cluster together, due to their similarity in bioactivity profiles (i.e. compounds which elicit neurochemical response more frequently interact at a complement of targets within protein families), as illustrated by the voltage-gated sodium channel Alpha subunit 1 and type III subunit (SCN1A and SCN3A) and the cholinergic muscarinic receptors 4 and 5 (CHRM4 and CHRM5) also corroborated in literature[56,57]. N.B.: Only drug targets that are clustering to at least three neurochemicals are listed here

**Table 1 Systematic analysis of the links between neurochemical components and predicted targets**

| Neurochemical component | Predicted target gene | Predicted target name | Reference link |
|---|---|---|---|
| Choline | ABCC2 | ATP-binding cassette subfamily C member 2 | Putative |
| | GRIN2A | Glutamate receptor ionotropic, NMDA 2A | Candidate[58] |
| | GRIN3A | Glutamate ionotropic receptor NMDA type subunit 3A | Putative |
| Dihydroxyphenylalanine | CHRM1 | Muscarinic acetylcholine receptor M1 | Confirmed[59] |
| | CHRM4 | Muscarinic acetylcholine receptor M4 | Confirmed[32] |
| | CHRM5 | Muscarinic acetylcholine receptor M5 | Confirmed[32] |
| Dihydroxyphenylethylene glycol | HTR1A | 5-hydroxytryptamine receptor 1A | Confirmed[60] |
| | HTR2C | 5-hydroxytryptamine receptor 2C | Confirmed[60] |
| | HTR5A | 5-hydroxytryptamine receptor 5A | Putative |
| Kynurenic acid | ABCC2 | Canalicular multispecific organic anion transporter 1 (ATP-binding subfamily C member 2) | Putative |
| | CAMK2A | Calcium/calmodulin dependent protein kinase II alpha | Candidate[61] |
| | CAMK2B | Calcium/calmodulin dependent protein kinase II beta | Candidate[61] |
| | GRIN1 | Glutamate receptor ionotropic, NMDA 1 | Putative |
| | GRIN2A | Glutamate ionotropic receptor NMDA type subunit 2A | Putative |
| | GRIN2B | Glutamate ionotropic receptor NMDA type subunit 2B | Putative |
| | GRIN2C | Glutamate ionotropic receptor NMDA type subunit 2C | Putative |
| | GRIN2D | Glutamate ionotropic receptor NMDA type subunit 2D | Putative |
| | HTR5A | 5-hydroxytryptamine receptor 5A | Putative |
| | PRKCZ | Protein kinase C zeta type | Putative |
| Oxytocin | DRD5 | Dopamine receptor D5 | Confirmed[62] |
| | OPRD1 | Opioid receptor delta 1 | Confirmed[31] |
| | OPRK1 | Opioid receptor kappa 1 | Confirmed[31] |
| Taurine | ADRA1B | Alpha-1B adrenergic receptor | Confirmed[30] |
| | DRD5 | Dopamine receptor D5 | Putative |
| Tryptophan | ABCC2 | Canalicular multispecific organic anion transporter 1 (ATP-binding subfamily C member 2) | Putative |

The table contains links between neurochemical components and protein target predictions with a 100% hit rate—i.e., every compound active at that neurotransmitter were predicted to interact with these targets. The links identified from bioactive compounds from the in silico target prediction protocol are shown along with any literature evidence. Putative links and pre-identified candidate links can be the focus of future neuro-biochemical studies

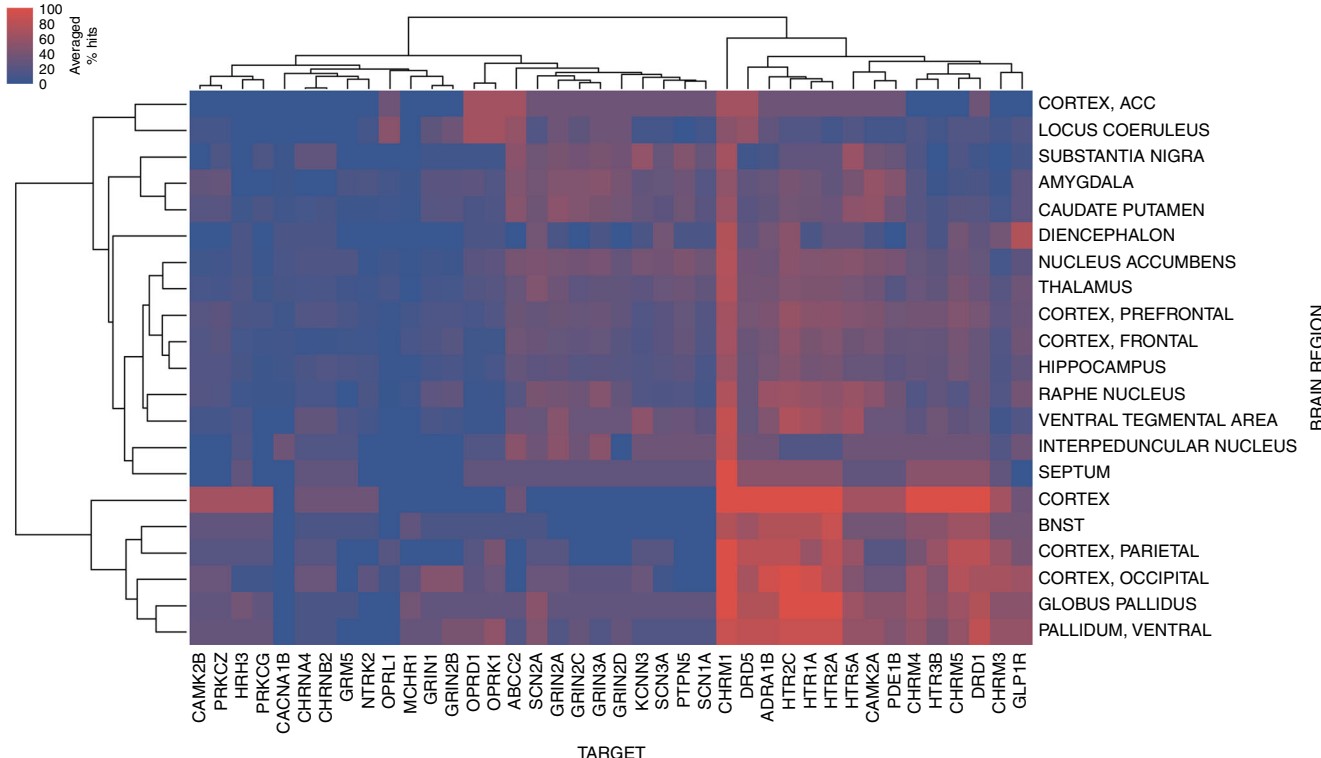

**Fig. 6** Systematic analysis of bioactivities against protein targets associated with neurochemical changes across brain regions. A larger percentage (red) indicates which protein targets are more often predicted to be targeted by drugs which change a given neurotransmitter level. N.B.: Only drug targets that are clustering to at least three neurochemical components are listed here

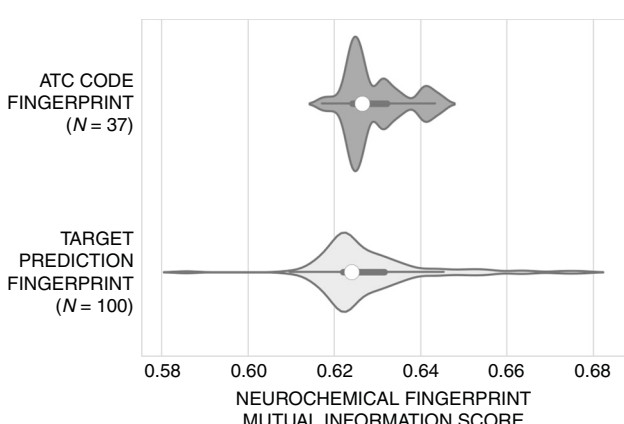

**Fig. 7** Mutual information between ATC codes and target prediction spaces with neurotransmitter response. The averaged (mean) mutual information across ATC codes and predicted targets is shown using kernel density estimation, with the median denoted by the white marker and the lower- and upper-quartile ranges shown using the thicker central lines. Overall, there is little difference between median mutual information scores with scores around ~0.630 and ~0.628, respectively. However, the underlying distributions are statistically different, with a two-sided Kolmogorov–Smirnov (KS) test p-value of 3.70e-4, and target prediction fingerprints comprise a larger standard deviation (0.010 vs. 0.007) in mutual information scores, spanning to scores over ~0.68. Hence, taken together we can also consider that there are specific targets which remain statistically more predictive of neurochemical response over ATC classes

variables on a per-ATC code basis) across the ATC classes (Fig. 8a) and the five protein targets providing the highest mutual information score, respectively, to identify which specific variables are most predictive of neurochemical response. It can be seen that the mutual information scores from the top 5 ATC classes comprise a bell-shaped distribution with averaged median values of ~0.07. In comparison, the top 5 informative predicted protein targets (Fig. 8b), possess higher mutual information compared to the aforementioned ATC classes, with a longer tail and a larger overall median of ~0.09. Hence the predicted protein targets possess higher mutual information with the neurochemical response of drugs than ATC classes. The most informative target is muscarinic cholinergic receptor 1 (CHRM1) based on the mutual information score, and although this was identified as an apparently promiscuous target in the previous analysis (since it was predicted to bind in an unspecfic manner to many different compounds that are active across regions and neurochemical components), hence indicates that there are *specific* interactions linked to CHRM1 that are predictive of specific neurochemical changes. Four of the highest ranked protein targets with respect to mutual information are linked with the serotonin receptor (HTR1A, HTR2C and HTR2A) or dopamine receptor (DRD5), which outlines how drugs binding to the group of protein targets linked with dopamine and serotonin (and their metabolites) produce more consistent neurochemical profiles, within certain brain regions at certain neurochemical components.

## Discussion

Current categories for the classification of psychiatric drugs are based on clinical consensus that is based on an earlier period of

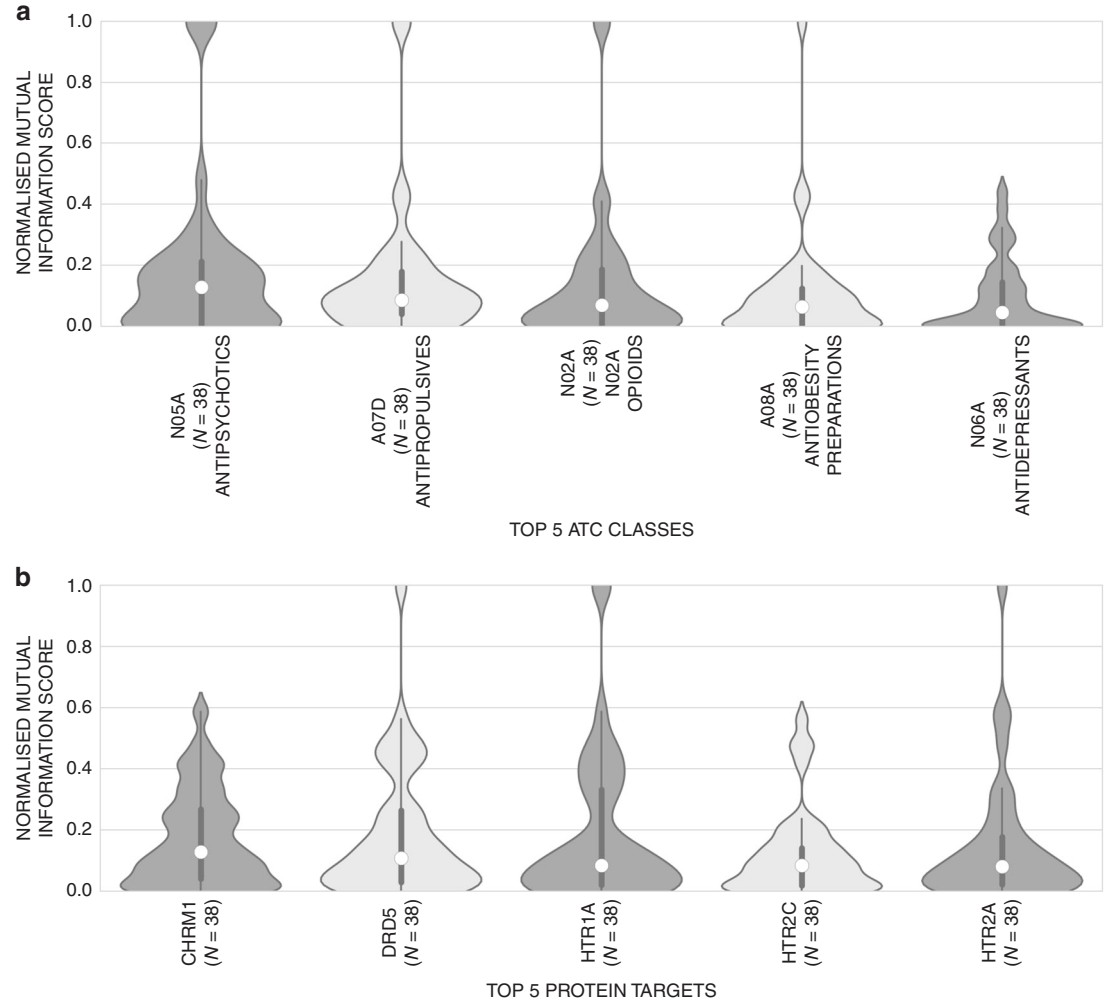

**Fig. 8** Top five most informative ATC classes based on neurotransmitter response patterns. **a** The top 5 informative codes ranked by median score are shown using kernel density estimation (with the lower- and upper-quartile ranges shown using the thicker central lines), which highlights which ATC classes are most correlated with neurochemical response. The distributions comprise a bell shape around scores of ~0.1. Classes are ranked left to right in descending order of median mutual information scores. **b** Top five most informative protein targets based on neurochemical response fingerprints. The distribution of the top 5 targets are shown using kernel density estimation (with lower- and upper-quartile ranges shown). and appear to be more informative than compared to the top 5 most informative ATC classes, since they comprise wider peaks around scores of ~0.5. Hence the most informative predicted protein targets appear to more predictive of neurochemical response then compared to the top ATC codes. Four of the top five targets are serotonin (HTR1A, HTR2C and HTR2A) or dopamine (DRD5) receptor related. Targets are ranked left to right in descending order of median mutual information scores

scientific understanding[34]. Here, we present a database built on multiscale neurochemical response patterns for therapeutic and experimental neuropsychiatric drugs that may pave the way for evidence-based classification strategies. The Syphad database assembled here will be essential for conducting studies in the field of neuropsychopharmacology as those studies rely on a precise understanding of the drug-induced neurochemical response patterns at systems level.

By applying chemoinformatics tools[20,21] we demonstrate the power of the Syphad database where we revealed links between molecular drug–target interactions and changes in neurotransmitter concentrations at connectome level. The complexity of brain diseases has led to recent interest in polypharmacology, which suggests that many effective drugs specially modulate multiple targets. In this respect, a drug that "hits" multiple sensitive nodes belonging to a network of neurotransmitter systems and interacting targets offers the potential for higher efficacy and

may limit drawbacks such as side effects generally arising from the use of a single-target drug or a combination of multiple drugs[35]. Our combined big data-chemoinformatics approach enhances the current understanding of the polypharmacology of neuropsychiatric drugs and contributes critically to the drug development and repurposing strategies. We further propose novel ways of modulating neurotransmitter levels by predicting target proteins. Based on those target predictions, our analyses suggest a mismatch between the current classification of neuropsychiatric drugs, spatiotemporal neurochemical response patterns at systems level, and drug–target interactions. In particular, our findings challenge the current view towards the dopaminergic system as a potential biomarker for psychiatric diseases.

Biomedical research has neglected many specific aspects of the health needs of women. This bias that is also reflected in *Syphad* as ~96% of all studies were conducted on male animals. This may

partly originate from the assumption that females, due to the cyclic reproductive hormones, are more variable than males. Sex-specific differences have been reported previously in local basal concentrations of neurotransmitters such as norepinephrine in thalamus[36], striatal dopamine[37] and acetylcholine in medial prefrontal cortex of rats[38], which may indicate differing responses to psychiatric drugs. Statistical comparison of normalized effect sizes with sex as a covariate was only possible for a very small subgroup, but did not show any significant differences between males and females. The skewness and sparsity of the data distribution limits the possibility to derive robust and reliable analytic results with respect to sex-specific differences and larger samples and test groups are required to obtain reproducible conclusions.

The drug classification system proposed in this work is built on region-specific multiscale neurochemical response patterns; however, it faces several limitations. Firstly, although our database derives from all published microdialysis measurements of drug-induced neurochemical alterations, the overall database has only a completeness of 2.6% when using the coarse (broad) ontology, as defined by the number of measured compound-brain region tuple data points divided by the total number of potential observable data points in the matrix. Over time the database will be enlarged by integrating new studies which will allow for a more precise compound classification. Secondly, the database contains an a priori skewness of data since almost 80% of all studies focus on monoaminergic systems, especially dopamine, while the most dominant excitatory and inhibitory neurotransmitters in the brain, glutamate and GABA, were only studied in 5% of the cases in total. This misbalance and overemphasis on dopaminergic neurotransmission may lead to an over-interpretation of the relevance of dopamine for pharmacotherapy of neuropsychiatric diseases. This effect may further suppress the identification of other transmitter systems for therapeutic purposes. Thirdly we do not know how well the neurochemical response patterns defined here for the rat brain translate to the human situation. However, rats provide a good model organism for testing the pharmacological action of drugs[39] and several microdialysis studies in rats showing changes in transmitter release were replicated in humans using positron emission tomography (PET)[40,41] or spectroscopy[42]. These similarities in rat and human brain on drug-induced neurochemical responses suggests construct validity of our database. Finally, the current content of our Syphad database relates to neurochemical responses to acute treatment with neuropsychiatric drugs, which may differ from clinical observations, since patients often receive chronic treatment for months and the drug effects only emerge after weeks of treatment. Hence, predictive validity is dependent on the inclusion of chronic dosing regimens, whereas acute-only results may be misleading for clinical interpretations. In particular, chronic administration of drugs such as ethanol[43], SSRI antidepressants[44] and antipsychotics[45] suggest that the effects may differ in dynamics and magnitude, sometimes even opposing to the acute drug effects. Therefore, particular care is advised in applying the database or the analytic findings of our study in a clinical context. Nonetheless, analysis of acute drug effects is not only a critical assessment tool for the potency of neuropsychiatric drugs in generating systemic effects but also to understand the brain function. Syphad facilitates such approaches by integrating the body of publications at large into a consistent framework that synergizes the cumulative knowledge of the past four decades of neuropsychopharmacology research.

In conclusion, Syphad is the first big data approach in the field of neuropsychopharmacology to systematically integrate existing information into a unified framework. Thereby, it sets a milestone towards evidence-based classification of CNS active drugs and thus, improves our understanding of the underlying neurobiological processes of neuropsychiatric diseases.

## Methods

**Search strategy.** The online portal of the National Library of Medicine [http://www.ncbi.nlm.nih.gov/pubmed/] including PubMed, PubMed Central and MEDLINE was used as the platform for literature research. A systematic screening of the original research articles published until 01.01.2016 was performed based on the keywords rat (AND) microdialysis (AND) (brain region (OR) neurotransmitter (OR) metabolite (OR) neuropeptide) (AND) (drug (OR) antidepressant (OR) anxiolytic (OR) psychostimulant (OR) sedative (OR) hypnotic (OR) antipsychotic (OR) neuroleptic.) The keyword neurotransmitter is a general representative in the search string which was replaced by the actual name and/or abbreviation of transmitters and metabolites (e.g., dopamine, glutamate, HVA etc). Additionally, separate searches were conducted substituting the keywords "drug", with the International Nonproprietary Name (INN) of all clinically approved and experimental neuropsychiatric drugs. If INN names were not assigned yet, USAN (United States Adopted Name) or BAN (British Approved Name) names were chosen. The full keyword-based search string was performed based on the 16,308 combinations of different brain regions, neurotransmitters and drugs designations and abbreviations (Supplementary Methods). In addition, the reference sections of identified papers as well as review and meta-analysis articles were then screened for further relevant citations.

**Study selection.** Reviewers, in pairs, independently screened titles and abstracts of articles and reviewed the full text of any title or abstract deemed potentially eligible by either reviewer. Reviewers resolved disagreements by discussion. Among these studies, only peer-reviewed original research articles in English language were chosen for data mining if they provided the absolute or relative change in neurotransmitter or metabolite concentrations within a brain region either numerically or in graphical manner. We excluded articles using animals other than rats. All selected studies were performed in outbred rats with no specific genotype or phenotype or provided data for a wild-type control group were included. Furthermore, animals did not receive any behavioural training prior to drug treatments. Abstracts and unpublished studies were not included. Authors were contacted if critical information was missing or only partially provided in their articles.

**Data extraction.** The following variables were extracted from the published studies by applying a structured template:

Biological variables: strain, sex, state of consciousness, i.e., awake or anesthetized (anaesthetic agent and the dosage), age, and number of animals used in each experiment.

Experimental procedure variables: coordinate of probe placement, sample time (min), flow rate (μl/min), membrane length (mm) of microdialysis probes, calcium concentration in perfusate (mM) and type of perfusate (e.g. Ringer solution), targeted brain region, neurochemical detection assay, route of drug administration, drug name and applied dose.

Experimental findings: drug dose effects (%) at time Ti, i.e., for a specific dose of the drug the absolute or relative changes of neurochemical concentrations within a brain region were obtained. The drug effects were normalized to the basal levels if absolute values were provided, in order to obtain relative changes.

**Quality assessment.** Two factors may have influenced the quality of the dataset: (1) differences in the drug nomenclature, in particular inconsistencies caused by reports using trade names of clinically approved drugs instead of INN or the International Union of Pure and Applied Chemistry (IUPAC) names. However, this issue was extremely rare and occurred in only two cases that allowed manual clustering of the drug names into the respective INN. (2) The accuracy, reliability and completeness of the microdialysis data. We addressed this matter by a twofold strategy. On the one hand, we conducted several sensitivity analyses (see below) to quantitatively evaluate the impact of missing effect modifiers, and on the other hand we conducted meta-analyses weighted by the number of animals used in each study. While we cannot verify the technical quality of conducted experiments, the number of animals provides a reliable measure to judge the statistical robustness of the findings of a study.

**Meta-analysis.** We conducted the meta-analysis of drug effects (%) using fixed effect model[36,44,46]: $\bar{x} = \frac{1}{N}\sum_{i=1}^{k} n_i x_i$, where $\bar{x}$ (effect size) represents the weighted average value as the weighted sum of the products of the drug effects $x_i$ obtained from each experiment $i$ and the number of animals used in that particular study $n_i$, and $N = \sum_{i=1}^{k} n_i$ denoting the total number of animals considered in the meta-analysis of the $k$ studies.

**Statistical analysis.** In order to assess the impact of inclusion of any partially non-independent study on the results, jackknife analyses were conducted iteratively. In other words, each partially non-independent study on a specific drug-dose-neurotransmitter-brain region combination was excluded and the weighted average

was recalculated. Subsequently, $\chi^2$ test or Fisher exact test was performed between the original and the leave-one-out recalculated statistics. Since no individual study skewed the overall statistics, the presented results are based on all studies. In addition, OFAT (one-factor-at-a-time) sensitivity analyses were performed a posteriori to ensure the robustness of the meta-analysis results with respect to the effect modifiers.

**Outcomes and effect modifiers**. The primary outcomes were matrices describing the peak changes of a specific neurotransmitter or metabolites (peak%baseline value) within distinct regions of rat brain for a specific drug–dose pairing. Inconsistencies in neuroanatomical nomenclature were avoided by using a previously developed[47] supervised machine learning technique to identify synonymous brain areas with respect to cytoarchitecture. A secondary outcome was the time-course of neurochemical alterations, characterized by the time-point at which the peak response occurred. Sex, age, strain, state of consciousness (i.e. use of anaesthesia), number of animals, dose of the drug, technical microdialysis parameters such as flow rate and calcium concentration of the perfusate, sampling time and length of the probe were considered as potential effect modifiers.

**Compound analysis based on experimental data**. Compounds in the dataset were annotated with 3rd level (pharmacological subgroup) ATC codes as retrieved from Drugbank[48], which describes the category a drug is assigned to based on current use (Supplementary Table 1). In all, 90 out of 258 clinically approved and experimental neuropsychiatric drugs had an available ATC mapping. Activity was defined as the minimum response recorded across all peak time points for each compound against a neurochemical component and brain region. A coarse-grained ontology was also used to employ a broad classification of brain regions, to reduce the number of brain regions, and to have more data per brain region (Supplementary Table 2). The overall database has a completeness of 2.6% when using the coarse (broad) ontology, as defined by the number of measured compound-brain region tuple data points divided by the total number of potential observable data points in the matrix.

**Data transformation**. RDKit [http://www.rdkit.org] was used to generate hashed circular chemical fingerprints[24] with a radius of 2 and 2048 bit length. The resulting bit array describes the presence and absence of chemical features for each of the drugs in the database, and is a common method to define the chemical similarity between two compounds[49].

For each drug–dose pairing, the primary outcomes (peak%baseline value) across neurotransmitter-brain region tuples were converted to bit array representations on a per-compound basis, to describe the neurochemical response patterns of each drug–dose pairing for comparison. Thus, the effect of different doses in neurochemical response patterns was explicitly integrated in the analysis. Each bit (corresponding to an individual experimentally confirmed neurotransmitter-brain region reading) was set via the following criteria; a bit was set to 1 if neurochemical response was increased above 100% and set to −1 for a decrease in response (below 100%).

For many drugs, the dose–response relationship is nonlinear. Therefore, dose equivalency considerations were omitted and instead machine learning classification algorithms were applied to characterize the impact of different drug doses (and indirectly receptor occupancy) in a hypothesis-free manner. Tanimoto similarity was calculated for the chemical fingerprints and for the neurochemical bit array representations between compounds within and across each ATC code using the Scipy http://www.scipy.org function spatial.distance.rogerstanimoto. For neurochemical response patterns this comparison only considered neurotransmitter-brain region tuples for which data was available for both compounds being compared.

**Clustering analysis**. Hierarchical clustering of the compounds in the database was performed using the matrix of compound and ATC codes and primary outcomes (peak%baseline value) within brain region-neurotransmitter tuples using the Seaborn [https://github.com/mwaskom/seaborn/tree/v0.8.0] *clustermap* function with the method set to *complete*, the metric set to *Euclidean*.

**In silico target prediction**. Next, in silico target deconvolution was performed, to annotate compounds with predicted targets using similarity relationships between the drugs in the database and identified ligands[20,21]. The algorithm output (flowchart outlined in Supplementary Fig. 3) is a probability for activity (binding) or inactivity (non-binding) on a per-compound basis across various protein targets. Although this method does not afford the prediction of the functional effects of compounds (i.e. activation or inhibition of a target), this analysis is useful since it enables the extrapolation of compound structure into bioactivity space and hence the identification of novel biological mechanism s to our analysis. This is particularly relevant, since there are incomplete bioactivity profiles for the full complement of protein targets expressed in the rat brain across all drugs in the database, and thus important proteins linked with biological activity are potentially unidentified. Four hundred and fifty-five drug-target bioactivity data points have been experimentally determined for the 258 drugs. Hence, if considering 100 protein targets are

expressed in the rat brain with an available bioactivity prediction model (full model details outlined in the next section), provides a completeness of only ~1.7% across 25,800 potential data points when using only the experimentally determined bioactivity matrix. By including in silico target predictions we can fill this (putative) bioactivity matrix completely, albeit with the knowledge that some of the predictions may not be accurate. This is in more detail described in the following.

To annotate the drugs in the database with their respective protein targets, we used the rat models available in PIDGIN version 2[50] on a per-compound bases. Previous benchmarking results have shown such in silico protocols perform with an average precision and recall of ~82% and ~83%, respectively, during fivefold cross validation[20], hence giving a reasonable likelihood that compounds predicted to bind a particular target will indeed bind to this protein, or set of proteins. We used a probability threshold of 0.5 to generate predictions in this work, where the predictions correlate for 319 of the 445 experimentally confirmed compound–target pairs for the drugs in our database (precision and recall of ~97% and ~84%, respectively). Importantly, the predictions from this analysis do not significantly contradict experimental results or significantly alter core findings when compared to an analysis consisting of entirely experimental biochemical data.

Predicted protein targets were filtered for those expressed in brain tissue as defined by the Human Protein Atlas[51], since region-specific genes have been shown to be conserved between both human and rat at the sequence and gene expression levels[52]. The following query was specified on the brain-specific proteome section of the resource: "tissue_specificity_rna:cerebral cortex;elevated AND sort_by:tissue specific score", providing 1437 targets with elevated expression in the brain compared to other organs (described from mRNA measurements and antibody-based protein experiments to identify the distribution of the brain-specific genes and their expression profiles compared to other tissue types[53]). Overall, 100 of the 515 (~19%) of the rat target models were retained after this filtering step (full list provided in Supplementary Table 3).

The proportion of drugs (eliciting neurochemical response) that were predicted to bind to a certain target within each neurotransmitter-brain region tuple (versus the predictions for all other drugs) were calculated, and used to identify correlations between brain location or neurotransmitter and molecular target spaces. The percentage of predicted drug–target interactions were aggregated by brain region, to annotate which bioactivities of drugs against protein targets lead to neurochemical component changes across brain regions. Percentages were also aggregated on a neurochemical component basis, to annotate the bioactivities of drugs against protein targets which lead to neurochemical component changes. The resulting matrices were filtered for display purposes for targets clustering to at least three brain regions or neurochemical components, respectively, and subjected to by-clustering using the Seaborn [https://github.com/mwaskom/seaborn/tree/v0.8.0] *clustermap* function with method set to *complete* and metric set to *Euclidean*.

**Mutual information analysis**. Drugs were annotated with predicted protein targets from the binary matrix of in silico target predictions. Next, drugs were annotated across the 38 available ATC codes with 1 for an annotation and 0 for no ATC class available. Finally, drugs were annotated using the matrix of neurochemical bit arrays across brain region and neurochemical components. The resulting ATC and protein target matrices were subjected to pairwise mutual information calculation against neurochemical bit arrays using the Scikit-learn function *sklearn.metrics.normalized_mutual_info_score*[54]. Drugs with missing neurochemical response patterns were removed per-pairwise comparison. This calculation results in a value between 0 (no mutual information) and 1 (perfect correlation). Scores were aggregated across ATC codes and targets and averaged to calculate the overall mutual information. Scores were also aggregated and ranked per-ATC code and per-predicted target to outline the top 5 informative features in either spaces.

**Reporting Summary**. Further information on research design is available in the Nature Research Reporting Summary linked to this article.

## Data availability

All data are available from the open-access database syphad [www.syphad.org]. The data used in the analysis is available for download as supplementary data to this manuscript and through Dryad repository[55]. A reporting summary is provided.

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

## Acknowledgements

This project has received funding from the European Union's Horizon 2020 research and innovation programme under grant agreement No 668863 (SyBil-AA). The work was also supported by the Bundesministerium für Bildung und Forschung (e:Med programme: FKZ: 01ZX1311A and FKZ: 01ZX1503). The authors wish to express their gratitude to Mr. Diego Sanchez for developing the MySQL database, inserting the data and IT administration.

## Author contributions

H.R.N. is the corresponding and lead author. H.R.N. and R.S. designed the study. Ö.D., L. E., S.F., B.G., H.-H.S., S.S. and H.R.N. collected data and performed meta-analysis. V.B. and H.R.N. pre-processed the data. L.M. and A.B. performed chemoinformatic analysis. L.M., A.B., R.S., N.K.L. and H.R.N. wrote and reviewed the paper.

## Additional information

**Competing interests:** The authors declare no competing interests.

