## [Peer Review File · Nature Communications]

Reviewers' comments:

Reviewer #1 (Remarks to the Author):

In this article, Noori and collaborators present a database of neurotransmitter response patterns in rat brain regions in response to administration of 258 neuropsychiatric drugs. The data in this database is extracted by manually curation of literature reports. Several analyses on the data are performed in order to support the view of a current discrepancy between drug therapeutic classification (ATC classes from WHO) and the actual neurochemical response patterns of the drugs. These analyses include the comparison of neurochemical response patterns for drug pairs within and across ATC classes, the clustering of ATC classes based on neurotransmitter-affected brain regions. Afterwards, they predict targets of the drugs and used this information to search for associations between molecular targets and neurotransmitters as well as brain region activities.

Overall, it is an interesting article not only for the newly generated database but also for the topic addressed. The data generated relies on previous published studies and suffer (as the authors also acknowledge) from incompleteness (data coverage is only 2.6%) and biases. This also limits the interpretability of the analysis and the robustness of the conclusions from the analyses. I see this as the mayor drawback of the article. Nevertheless, the article is a pioneer study in a far from complete understood area of brain pharmacology and therefore, I am positive about it. Below are more general as well as specific comments on the study that might help to improve the clarity of the manuscript.

Comments.

As a general trend, the article avoids details and technical descriptions on the main manuscript. Both the data as well as the analysis are described in general terms without (most of the time) reference to the Method section or Supplementary information where the technical information is described. This left the reader that does not read the Method section before reading the main text with an incomplete understanding of the analysis. An overview initial figure illustrating the data that is used for the analyses, how the fingerprints are constructed and an illustration of performed analyses will be of great help for the reader. In addition, references to the Supplement or to the Methods section should be placed in the main manuscript so that the interested reader of the location of this information. To illustrate this point, in the main text it is not described how the "neurochemical response profiles" are constructed. It is only mentioned that this response originated from in vivo microdialysis studies. There is also not a reference in place to the Methods section where this information is described.

It is often not stated the data that is used for each analysis and more importantly the measurements that are taken in order to make sure that the results are robust. With data coverage of 2.6%, the results are likely to change when this coverage is complete. For example, when comparing fingerprint profiles of neurotransmitters for different drugs, the similarity of the profiles can change substantially when only information about 2 neurotransmitter or information for the full set (59) of neurotransmitters is available. Along the same line, the matrix data used for Figure 4 is sparse. The authors select data with 50% coverage. However, the bias on the tissues studied might skew the analysis, not showing the real similarity of the activity of drugs across tissues. The underlying data for this figure should be provided. How many tissues are included in the analysis?. Another example is the information on drug-target prediction, which is not explicit and should be provided as Supplementary information.

Another example of important information missed on the main text is the reason for selection of only 100 targets for the molecular analysis. This information is only on the Method section. I would suggest the author to include the motivation for the target selection on the main manuscript.

As a summary of the comments above, I recommend the authors to provide the information used

for every analysis and assess the robustness of the results.

The analysis of the associations between drug targets and neurotransmitters is interesting and an important potential application of the analysis. To support the results, the authors provide some confirmatory examples extracted from the literature (e.g. ADRA1B activation potentiates taurine response mediated by PKC., OPRD1 modulates oxytocin levels). However, there is not a systematic analysis of all predicted associations or additional experimental confirmation for some new relations. Since the data used for the analysis is predicted information, for a more robust assessment and in order to enlighten the study with new findings, it would be interesting to confirm some of the unknown associations experimentally.

The neurochemical response profile for the group intra-ATC class should be the same in Figure 1 and Figure 2 (ALL). However, the profiles do not look completely similar. Why ?

In Figure 3, the full name of the compounds should be given, if possible in the same figure (replacing the current short names) or alternatively pointing to the abbreviation list with all the short-long name correspondences in the supplement. A similar comment for Figure 5: for a better readability, I would recommend to display the full name of the compounds in the graph.

In page 8, line 158, a reference to Supplementary Figure 1 is made in relation to chemical structures. However, this figure illustrates the *in silico* target deconvolution protocol.

Discussion section:

I find that some conclusions on the discussion section are overstated. In particular these two: "the analysis lead to the identification of novel ways of modulating neurotransmitter levels" and "identify causal relationships between molecular drug-target interactions and changes in neurotransmitter concentrations at a connectome level". In my view, the analyses as they stand now suggest, rather than identify, novel ways of modulating neurotransmitter levels and possible relations between molecular drug-target interactions and changes in neurotransmitter concentrations.

I miss a paragraph discussing the translation of experimental observations of the rat species to the human organism and to what extent the results of the analysis with rat data reflect the drug effects in human organism.

Although a comment on data completeness in the discussion is made and acknowledged, the authors should also acknowledge the limitations of the data completeness to extract robust results.

I have problems understanding the statement of "the database is built on spatiotemporal response patterns...and follows the principles of polypharmacology". As far as I can see, this database does not include drug-target information. What do the authors mean by "polypharmacology" in the database?

Page 18, line 375: Inconsistencies in neuroanatomical nomenclature inconsistencies.
"Inconsistencies" is redundant.

Comments on the Database:

Although the database is central on the manuscript, the content of the database and the possibilities of it are not clear. The authors performed analysis with the upregulation and downregulation of certain neurotransmitters, however it is not clear where this information appears on the output of database search. In the website of the database, a help page with details

on the parameter retrievable as well as examples is recommended. As far as I can see, although the information is freely accessible it is not downloadable. This limits the potential use of the database for other researchers. Some typos (in reset search parameters)

Reviewer #2 (Remarks to the Author):

In this article the authors present a detailed analysis of available neurochemical data for a large number of compounds used in the treatment of psychiatric disorders or with targets which are linked to psychopharmacology. The authors used a number of different methods to interrogate this large database and evaluate how traditional drug classification methods compare to the in vivo neurochemical changes induced by these drugs across different brain areas. The main findings from the analysis are that the classification of agents based on the ATC method (using their main therapeutic indication) does not compare well with their neurochemical effects. Given that the neurochemical effects are more likely to be related the behavioural effects of these drugs, this would suggest that these traditional drug classification methods are limited. The manuscript represents a very large body of work which illustrates the value of a big data approach to trying to better understand the relationship between a drugs pharmacological targets and understanding how it can affect behaviour. The work highlights how poorly we currently understand why certain drugs are effective in different clinical conditions and perhaps also illustrates the limitations of our current knowledge. I am not an expert in the methods used here and found some of the text difficult to follow but generally the figures were clear and the main findings will certainly be of interest to both specialist and non-specialist readers. The database on which this work is based is also a very valuable resource for other researchers in this field and, as more mining of the data is undertaken, will likely reveal further insights as well as preventing additional, unnecessary animal experiments which is important when considering the 3Rs. There are some specific comments which I think would be useful to consider from the perspective of clarity to a non-specialist reader and also a couple of areas where I think a broader consideration of current ideas in this field could be incorporated.

1. The manuscript does not mention the Neuroscience-based nomenclature (www.nbn2.com/) which I think is quite relevant to the discussions here particularly in relation to drug classification. More broadly, I think it would be worth including a more general introduction which takes into consideration the fact that many drugs are used for more than one indication as defined in DSM. The most obvious is the SSRIs and their use in MDD and anxiety disorders but other examples exist. For example, Page 4, Ln 72-75, this statement relating to drug classification does not really consider the clinical picture where drugs classed as an antidepressant, anxiolytic or antipsychotic may be given to patients with a primary diagnosis in any one of these areas. Patients may also receive several treatments in combination and this needs to be considered here to reflect the clinical scenario.

2. The focus here is on the acute effects of drugs but most psychiatric treatments are used chronically and may well have different effects following longer term use than their initial acute effects. For example, animal studies show that the monoamine re-uptake inhibitors and MAOI induce downregulation of autoreceptors in the noradrenergic and serotonergic system. They also induce neurotrophic changes. The typical antipsychotics also induce adaptive changes in dopamine receptors. The clinical benefits of these drugs may also be delayed and so their clinical efficacy may be less about their acute neurochemical effects and more about the changes they induce in the brain. I am not suggesting that these data are included but I think some consideration of this should be included.

3. A major concern with regard to data obtained from animal studies is how well the dose used compares to receptor occupancy and the level of occupancy required for clinical efficacy. Although dose is included in the database, I was not clear how this was factored into the analysis and subsequent conclusions. If a drug is given at a high enough dose it may well effect transmitter systems which would not be altered at lower doses. I may have missed this in the methods but

was there consideration given to dose equivalency in the analysis when deciding which data to use in the correlations and other comparative studies?

4. There seems to me to be some repetition between the introduction and discussion which is unnecessary and could be revised to make room for consideration of some of the other points raised in this review if needed.

5. In the final paragraph of the discussion the authors consider sex differences in terms of the bias towards studies in males, where data was available for both sexes, how well did this concur? If this discussion is included, I think this would a useful comparison to include. Although I agree that sex differences are important, evidence to support this would be useful if available even for only a small number of drugs.

Submission of revised version of „Neurochemical Fingerprints of Neuropsychiatric Drugs“ (NCOMMS-18-16325-T)

Response to Reviewers

Reviewer #1:

R1.1. Overall, it is an interesting article not only for the newly generated database but also for the topic addressed. The data generated relies on previous published studies and suffer (as the authors also acknowledge) from incompleteness (data coverage is only 2.6%) and biases. This also limits the interpretability of the analysis and the robustness of the conclusions from the analyses. I see this as the mayor drawback of the article. Nevertheless, the article is a pioneer study in a far from complete understood area of brain pharmacology and therefore, I am positive about it. Below are more general as well as specific comments on the study that might help to improve the clarity of the manuscript.

A: We thank the reviewer sincerely for the time and energy and constructive feedbacks that were offered to improve our manuscript. We have embraced all comments and suggestions in the revised manuscript. In general, the main text now contains substantially more information and detail, overview figure was added and additional analyses with respect to robustness of findings are conducted. The whole database as an excel downloadable file is now provided as supplementary data file to the paper.

R1.2. As a general trend, the article avoids details and technical descriptions on the main manuscript. Both the data as well as the analysis are described in general terms without (most of the time) reference to the Method section or Supplementary information where the technical information is described. This

left the reader that does not read the Method section before reading the main text with an incomplete understanding of the analysis.

An overview initial figure illustrating the data that is used for the analyses, how the fingerprints are constructed and an illustration of performed analyses will be of great help for the reader.

A: We now provide an overview illustration at beginning of results section (Figure 1) that summarizes the process of data mining, fingerprint construction and analyses.

R1.3. In addition, references to the Supplement or to the Methods section should be placed in the main manuscript so that the interested reader of the location of this information. To illustrate this point, in the main text it is not described how the “neurochemical response profiles” are constructed. It is only mentioned that this response originated from in vivo microdialysis studies. There is also not a reference in place to the Methods section where this information is described.

A: We thank the reviewer for bringing this issue to our attention. We have now systematically added references to methods section within the main text to guide the reader to the location of the information. Furthermore, we provide additional information within the main text as well to improve the clarity of the presented content to avoid the need for going back and forth within the paper.

R1.4. It is often not stated the data that is used for each analysis and more importantly the measurements that are taken in order to make sure that the results are robust. With data coverage of 2.6%, the results are likely to change when this coverage is complete. For example, when comparing fingerprint profiles of neurotransmitters for different drugs, the similarity of the profiles can change substantially when only information about 2 neurotransmitter or information for the full set (59) of neurotransmitters is available. Along the same line, the matrix data used for Figure 4 is sparse. The authors select data with 50% coverage. However, the bias on the tissues studied might skew the analysis, not showing the real similarity of the activity of drugs across tissues. The underlying data for this figure should be provided. How many tissues are included in the analysis?. Another example is the information on drug-target prediction, which is not explicit and should be provided as Supplementary information.

A: We now provide detailed information of the input data for each analysis and for each figure. In particular, we provide the number of animals and studies used. Furthermore, we have now conducted additional sensitivity analyses in order to investigate the robustness of our findings.

R1.5. Another example of important information missed on the main text is the reason for selection of

only 100 targets for the molecular analysis. This information is only on the Method section. I would suggest the author to include the motivation for the target selection on the main manuscript.

A: As suggested by the reviewer, we now include the motivation and detailed information for target selection in the main manuscript.

R1.6. As a summary of the comments above, I recommend the authors to provide the information used for every analysis and assess the robustness of the results.

A: We agree with the referee to 100% and have conducted sensitivity analyses to assess the robustness of the results. Moreover, the information related to each analysis is provided at place in the main text. In particular, we now report sensitivity analyses at two levels. The sensitivity of meta-analyses to biological covariates are reported in detail. Moreover, the robustness of chemoinformatic analyses was investigated.

R1.7. The analysis of the associations between drug targets and neurotransmitters is interested and an important potential application of the analysis. To support the results, the authors provide some confirmatory examples extracted from the literature (e.g. ADRA1B activation potentiates taurine response mediated by PKC..., OPRD1 modulates oxytocin levels). However, there is not a systematic analysis of all predicted associations or additional experimental confirmation for some new relations. Since the data used for the analysis is predicted information, for a more robust assessment and in order to enlighten the study with new finding, it would be interesting to confirm some of the unknown associations experimentally.

A: Indeed, we conducted a systematic analysis of all predicted associations but all findings with exception of two examples were only reported in the figures. In the revised version, we now report accurately and in detail the most robust findings of the systematic analysis as an outlook for future investigations.

R1.8. The neurochemical response profile for the group intra-ATC class should be the same in Figure 1 and Figure 2 (ALL). However, the profiles do not look completely similar. Why ?

A: We thank the reviewer for bringing this to our attention. This was an unfortunate and bad choice in denotation of the plots. ALL category in figure 2 shows the filtered ATC classifications in the plot (i.e. does not represent global distribution). We have now changed the term and use more precise wording

instead. Furthermore, we have changed the presentation of figure 2 into Whisker plots. Complementary the recommendations by the referee to include the sensitivity analyses to the results, we believe that the Whisker plot presentation can reflect the robustness of results more effectively. Hence, while it was not mentioned in the review, we took the liberty to improve the style of the paper in the gist of referee's comments and improved the figures as well.

R1.9. In Figure 3, the full name of the compounds should be given, if possible in the same figure (replacing the current short names) or alternatively pointing to the abbreviation list with all the short-long name correspondences in the supplement. A similar comment for Figure 5: for a better readability, I would recommend to display the full name of the compounds in the graph.

A: As suggested by the reviewer, both figures now contain full name of compounds instead of abbreviations.

R1.10. In page 8, line 158, a reference to Supplementary Figure 1 is made in relation to chemical structures. However, this figure illustrates the in silico target deconvolution protocol.

A: This error has now been corrected.

R1.11. I find that some conclusions on the discussion section are overstated. In particular these two: “the analysis lead to the identification of novel ways of modulating neurotransmitter levels” and “ identify causal relationships between molecular drug-target interactions and changes in neurotransmitter concentrations at a connectome level”. In my view, the analyses as they stand now suggest, rather than identify, novel ways of modulating neurotransmitter levels and possible relations between molecular drug-target interactions and changes in neurotransmitter concentrations.

A: We have now toned down and rephrased both statements as suggested by the referee.

R1.12. I miss a paragraph discussing the translation of experimental observations of the rat species to the human organism and to what extent the results of the analysis with rat data reflect the drug effects in human organism.

A: We agree with the reviewer that this is an important aspect that was unfortunately missing in the original version. Thus, we have now included a discussion on the translation of rat findings to humans and the overall reach of our findings for clinical cases.

R1.13. Although a comment on data completeness in the discussion is made and acknowledged, the authors should also acknowledge the limitations of the data completeness to extract robust results.

A: As suggested by the reviewer, we now discuss the sensitivity analysis and robustness issues associated with analysis of sparse datasets such as ours in the discussion section.

R1.14. I have problems understanding the statement of “the database is built on spatiotemporal response patterns...and follows the principles of polypharmacology”. As far as I can see, this database does not include drug-target information. What do the authors mean by "polypharmacology" in the database?

A: Our intention was to express that drugs modulate multiple targets and that the database is not restricted to the „one drug“ „one transmitter“ but encompasses all possible affected systems. We agree however, that our initial phrasing is confusing. Thus, we have rephrased the statement to avoid any ambiguity and confusion.

R1.15. Page 18, line 375: Inconsistencies in neuroanatomical nomenclature inconsistencies. “Inconsistencies” is redundant.

A: Corrected.

R1.16. Although the database is central on the manuscript, the content of the database and the possibilities of it are not clear. The authors performed analysis with the upregulation and downregulation of certain neurotransmitters, however it is not clear where this information appears on the output of database search. In the website of the database, a help page with details on the parameter retrievable as well as examples is recommended. As far as I can see, although the information is freely accessible it is not downloadable. This limits the potential use of the database for other researchers. Some typos (in reset serach parameters)

A: We now provide substantially more information on the database in the results section. We have also added a how-to guide and remarks for the online database on the search page and corrected the typos. Moreover, we now provide a single excel file as downloadable supplementary data of this paper to Nature Communications, which contains the entire database.

Reviewer #2:

We are very grateful to the reviewer for her/his appreciation of our work and for the valuable comments to improve our manuscript.

R2.1. The manuscript does not mention the Neuroscience-based nomenclature (www.nbn2.com/) which I think is quite relevant to the discussions here particularly in relation to drug classification. More broadly, I think it would be worth including a more general introduction which takes into consideration the fact that many drugs are used for more than one indication as defined in DSM. The most obvious is the SSRIs and their use in MDD and anxiety disorders but other examples exist. For example, Page 4, In 72-75, this statement relating to drug classification does not really consider the clinical picture where drugs classed as an antidepressant, anxiolytic or antipsychotic may be given to patients with a primary diagnosis in any one of these areas. Patients may also receive several treatments in combination and this needs to be considered here to reflect the clinical scenario.

A: We thank the reviewer for bringing this to our attention. We have now significantly extended the introduction and address the issues accompanied with indication-based nomenclature with a few examples as recommended by the referee. Moreover, we discuss NbN in the introduction section as a necessary multimodal pharmacologically driven classification system that revisited the 60 year old concepts. Appropriate citations are added to the main text.

R2.2. The focus here is on the acute effects of drugs but most psychiatric treatments are used chronically and may well have different effects following longer term use than their initial acute effects. For example, animal studies show that the monoamine re-uptake inhibitors and MAOI induce downregulation of autoreceptors in the noradrenergic and serotonergic system. They also induce neurotrophic changes. The typical antipsychotics also induce adaptive changes in dopamine receptors. The clinical benefits of these drugs may also be delayed and so their clinical efficacy may be less about their acute neurochemical effects and more about the changes they induce in the brain. I am not suggesting that these data are included but I think some consideration of this should be included.

A: We agree with the reviewer completely. In the revised version, we discuss this matter in detail and with examples. In particular, the discussion addresses now the importance of chronic drug administration in clinical context, differences in magnitude and dynamics of drug responses between acute and chronic drug exposure and advise the reader to treat the findings of our study with care if (s)he intends to apply them in clinical context.

R2.3. A major concern with regard to data obtained from animal studies is how well the dose used compares to receptor occupancy and the level of occupancy required for clinical efficacy. Although dose is included in the database, I was not clear how this was factored into the analysis and subsequent conclusions. If a drug is given at a high enough dose it may well effect transmitter systems which would not be altered at lower doses. I may have missed this in the methods but was there consideration given to dose equivalency in the analysis when deciding which data to use in the correlations and other comparative studies?

A: In the generation of neurochemical fingerprints that form the elementary components of our analysis, each fingerprint was associated uniquely to a (drug, dose) pairing. Thus, the effect of different doses in neurochemical response patterns was explicitly integrated in the analysis. Since the dose-response relationship was often nonlinear, we applied machine learning classification algorithms, instead of dose equivalency assumptions to characterize the impact of different doses on the comparative studies. We emphasized this crucial aspect in the revised version of the manuscript and made the relevant facts easier accessible.

R2. 4. There seems to me to be some repetition between the introduction and discussion which is unnecessary and could be revised to make room for consideration of some of the other points raised in this review if needed.

A: In full agreement with the reviewer's suggestion, we have now substantially rewritten the discussion section, avoided redundancies and made room to address the comments made in this review by both referees.

R2.5. In the final paragraph of the discussion the authors consider sex differences in terms of the bias towards studies in males, where data was available for both sexes, how well did this concur? If this discussion is included, I think this would a useful comparison to include. Although I agree that sex differences are important, evidence to support this would be useful if available even for only a small number of drugs.

A: The meta- and sensitivity analyses in our database did not show any sex-specific differences. Only 5 drugs allowed a statistical comparison of mean effects with sex as a covariate. We now report the findings in results section (Database sensitivity analysis). Furthermore, as suggested by the referee, we

now provide some evidence and additional references supporting the discussion section, which has been rephrased significantly to embrace all above comments smoothly.

REVIEWERS' COMMENTS:

Reviewer #1 (Remarks to the Author):

The authors have addressed all my points and have no other remarks to the manuscript. The manuscript has improved substantially. The new figure 1 is illustrative. The sensitivity analyses to assess the robustness of the analyses reinforce the results. I congratulate the authors for the work.

Reviewer #2 (Remarks to the Author):

The authors have carefully revised their manuscript in line with the recommendations for the reviewers and the work is now much clearer. I have no further comments.

Response to Reviewers

Reviewer #1 (Remarks to the Author): The authors have addressed all my points and have no other remarks to the manuscript. The manuscript has improved substantially. The new figure 1 is illustrative. The sensitivity analyses to assess the robustness of the analyses reinforce the results. I congratulate the authors for the work.

Reviewer #2 (Remarks to the Author): The authors have carefully revised their manuscript in line with the recommendations for the reviewers and the work is now much clearer. I have no further comments.

A: We thank both reviewers for the time and effort spent on our manuscript. We are delighted that the revised manuscript addressed their previous concerns satisfactory